# Prognostic Significance of PD-L1 Expression in Gastric Cancer Patients with Peritoneal Metastasis

**DOI:** 10.3390/biomedicines11072003

**Published:** 2023-07-15

**Authors:** Xiao-Jiang Chen, Cheng-Zhi Wei, Jun Lin, Ruo-Peng Zhang, Guo-Ming Chen, Yuan-Fang Li, Run-Cong Nie, Yong-Ming Chen

**Affiliations:** State Key Laboratory of Oncology in South China, Department of Gastric Surgery, Collaborative Innovation Center for Cancer Medicine, Sun Yat-sen University Cancer Center, No.651 Dongfeng Road East, Guangzhou 510060, China; chenxiaoj1@sysucc.org.cn (X.-J.C.); weicz@sysucc.org.cn (C.-Z.W.); linjun@sysucc.org.cn (J.L.); zhangrp@sysucc.org.cn (R.-P.Z.); chengm@sysucc.org.cn (G.-M.C.); liyuanf@sysucc.org.cn (Y.-F.L.)

**Keywords:** gastric cancer, peritoneal metastasis, PD-L1, prognosis

## Abstract

Background: Recently, many studies have explored the relationship between the expression of programmed death ligand 1 (PD-L1) and prognosis in gastric cancer, but there is still controversy. Additionally, few studies have specifically investigated the expression of PD-L1 in patients with peritoneal metastasis. Methods: Immunohistochemistry was used to analyze the expression of PD-L1 in gastric cancer patients with peritoneal metastasis. The combined positive score (CPS) was calculated to evaluate the expression of PD-L1, and the clinicopathological data were analyzed to explore prognostic significance. Results: In total, 147 gastric cancer patients with peritoneal metastasis were enrolled. The negative PD-L1 expression was defined as a CPS < 1, and high PD-L1 expression was defined as a CPS ≥ 10. PD-L1 expression with CPS ≥ 1 and CPS-negative was detected in 67 (45.58%) and 80 (54.42%) patients, respectively. High PD-L1 expression at PD-L1 CPS ≥ 10 was detected in 21(14.29%) patients. The median overall survival (OS) was 18.53 months in the CPS < 10 group and 27.00 months in the CPS ≥ 10 group; the OS difference between the two groups was significant (*p* = 0.015). Multivariate analysis demonstrated that a poor Eastern Cooperative Oncology Group performance score (ECOG PS) (*p* = 0.002) and severe peritoneal metastasis (*p* = 0.033) were significantly associated with poor survival, while palliative chemotherapy (*p* = 0.002) and high PD-L1 expression (*p* = 0.008) were independent and significantly favorable prognostic factors. Conclusions: Our study demonstrated that PD-L1 expression was widely presented in gastric cancer patients with peritoneal metastasis, while a CPS no less than 10 predicted better prognosis.

## 1. Introduction

Gastric cancer remains the fifth most common cancer and the fourth leading cause of cancer death all over the world [1]. In China, the morbidity and mortality of gastric cancer were only second to lung cancer [2]. More than 80% of patients are diagnosed with advanced gastric cancer at their first visit [3]. Although the patient’s response to chemotherapy [4], radiotherapy [5], and molecular targeted therapy [6] has improved significantly, the prognosis of metastatic gastric cancer remains unsatisfactory.

When patients with gastric cancer present emergency conditions, such as bleeding, obstruction, and perforation, it often indicates the advanced stage and poor prognosis [7,8]. Peritoneal metastasis was usually observed during surgical treatment for the above emergency conditions [9,10]. Peritoneal metastasis is the most common pattern of metastasis and cause of death in patients with gastric cancer [11,12]. The main mechanism is believed to be the infiltration of tumor cells into the serous layer to form free cells and further complete the colonization and metastasis [13,14].

The programmed death ligand 1 (PD-L1) is the ligand to the programmed cell death protein-1 (PD-1) receptor, which is mainly expressed in immune and tumor cells [15]. Currently, checkpoint pathway blockades of PD1/PD-L1 have been a highly promising immunotherapy by activating T lymphocytes and enhancing antitumor immunity, thus leading to an impressive outcome in patients with gastric cancer [16,17,18,19]. Numerous clinical trials have made immunotherapy available to patients in every line of therapy. The combination therapy of nivolumab plus chemotherapy achieved a clinically significant overall survival (OS) benefit in the first-line setting in all advanced esophageal and gastric adenocarcinoma patients with approval in Europe, the USA, and other countries [18]. The combination of pembrolizumab plus chemotherapy was approved for patients with esophageal cancer and Siewert-I gastroesophageal junction cancer both in Europe and the USA [20]. As for patients with Her2 overexpression tumors, the combination of trastuzumab, pembrolizumab, and chemotherapy demonstrated superior results in response and was approved as a first-line therapy option in the USA [21]. Even in the third-line setting, nivolumab prolonged OS compared to the placebo and was approved in Japan [16]. Furthermore, pembrolizumab significantly prolonged the duration of the reaction, resulting in approval for patients with PD-L1 CPS ≥ 1 tumors in the USA [17]. The approval of immune checkpoint inhibitors has enhanced the current treatment options and has provided a viable, personalized treatment option for advanced gastric cancer patients.

At present, numerous studies have explored the relationship between the expression of PD-L1 and prognosis in gastric cancer, which is still controversial [22,23,24,25,26]. Boger et al. [22] investigated the expression of PD-L1 and PD-1 in a large and well characterized gastric cancer cohort of Caucasian patients; the results showed that high PD-L1/PD-1 expression was associated with a significantly better patient outcome, and PD-L1 turned out to be an independent survival prognosticator. Xing et al. [26] analyzed PD-L1 expression in 1014 patients with gastric cancer, and the results indicated that high PD-L1 expression exhibited better survival. However, Eto et al. [24] reported that the expression of PD-L1 was detected in patients with gastric cancer that underwent curative gastrectomy and found that the positive PD-L1 expression patients tend to have lower overall survival than the negative PD-L1 expression patients. Nevertheless, the relationship between PD-L1 expression and prognosis in patients with gastric cancer with peritoneal metastasis remains unclear.

Therefore, the aim of our study was to analyze clinicopathological features and quantitatively detect the expression of PD-L1 in gastric cancer patients with peritoneal metastasis and to further investigate the relationship between the expression of PD-L1 and prognostic significance.

## 2. Materials and Methods

### 2.1. Patients

Patients were eligible if they were pathologically diagnosed with gastric adenocarcinoma with peritoneal metastasis and underwent palliative gastrectomy in Sun Yat-sen University Cancer Center between January 2000 and December 2015. Patients were not treated with chemotherapy, radiotherapy, or immunotherapy at the time of enrollment. The postoperative palliative chemotherapy regimen was unified as platinum and fluorouracil, and none of the patients received immunotherapy in the course of disease treatment. The dose of chemotherapy for each patient was determined by the treating oncologists, as per the National Comprehensive Cancer Network (NCCN) guidelines. Well-preserved pathological sections were available for all enrolled patients.

This study was approved by the institutional review committee of Sun Yat-sen University Cancer Center. The authenticity of this study was verified by uploading the raw data to the Research Data Deposit public platform (www.researchdata.org.cn, accessed on 15 May 2023).

### 2.2. Data Collection

We reviewed the clinicopathologic characteristics for all enrolled patients. The characteristics included the gender (female, male), age (<60, ≥60 years old), Eastern Cooperative Oncology Group performance score (ECOG PS) (0/1, 2/3), tumor localization (upper, middle, lower), pathological differentiation (moderately, poorly), Lauren classification (diffuse, intestinal, mixed), primary tumor stage (T stage), regional lymph nodes stage (N stage), degree of peritoneal metastasis, and postoperative chemotherapy treatment.

The TNM staging classification was performed according to the American Joint Committee on Cancer (AJCC) for Carcinoma of the Stomach (8th ed., 2017). The degree of peritoneal metastasis is classified according to the first English edition of the Japanese classification of gastric carcinoma as follows: P0, no peritoneal metastasis; P1, disseminating metastasis to the region directly adjacent to the peritoneum of the stomach (above the transverse colon, including the greater omentum); P2, several scattered metastases to the distant peritoneum and ovarian metastasis alone; P3, numerous metastases to the distant peritoneum [27].

The overall survival (OS) was calculated as the period from the diagnosis of gastric cancer with peritoneal metastasis to death, and the progression-free survival (PFS) was defined as the period from the diagnosis of gastric cancer with peritoneal metastasis to tumor progression or death, whichever came first.

Follow-up was acquired through telephone and outpatient information. Follow-up was recommended once every 3–6 months in the first 2 years, followed by once every 6–12 months until 5 years after palliative surgery. All follow-up assessments were completed by June 2020, and the median follow-up time was 15.77 months (range, 1.00–152.00 months).

### 2.3. Immunohistochemistry and Scoring of PD-L1 Expression

Immunohistochemical staining of PD-L1 expression was performed using the Dako 22C3 pharmDx assay (Agilent Technologies, Carpinteria, USA) on tissue sections of primary tumor specimens removed by palliative gastrectomy for all patients. PD-L1 expression was evaluated in both tumor and immune cells by two pathologists. The combined positive score (CPS) was calculated by dividing the number of PD-L1 stained cells (tumor cells and immune cells) by the total number of viable tumor cells and by multiplying the value by 100. The PD-L1 CPS is currently recognized as a cost-effective screening tool that is easy to apply clinically [28]. The PD-L1 negative was defined as a CPS < 1, and high PD-L1 expression was defined as a CPS ≥ 10.

### 2.4. Statistical Analysis

All the statistical analyses were performed using the IBM SPSS version 25.0 (SPSS, Chicago, IL, USA). Chi square tests were used to compare categorical variables. Univariate and multivariate analyses for overall survival were performed using Cox’s regression analysis. Variables with a *p* value < 0.05 in the univariate analysis were entered into a multivariate analysis using Cox proportional hazard regression models. The hazard ratio (HR) and 95% confidence interval (CI) were used to estimate the survival predictor.

Kaplan–Meier survival curves with log-rank testing were performed to compare the survival benefits. Statistical significance was set at *p* < 0.05, and statistical analyses were performed using R version 4.2.2 (R Foundation for Statistical Computing, Vienna, Austria).

## 3. Results

### 3.1. Patient Baseline Characteristics

In this study, a total of 147 gastric cancer patients with peritoneal metastasis were enrolled, including 89 (60.5%) males and 58 (39.5%) females. The baseline characteristics for all patients are shown in Table 1. The median age at diagnosis was 54 years (range: 19–84 years). The ECOG PS of 114 (77.6%) cases were 0 or 1, and only 33 (22.4%) cases were 2 or 3. A total of 45 (30.6%) cases were located at the upper part of the stomach, 37 (25.2%) cases were located at the middle part, and 62 (44.2%) cases were located at the lower part. Pathology results for all patients were adenocarcinoma, while 15 (10.2%) cases were moderately differentiated and 132 (89.8%) cases were poorly differentiated. For the Lauren classification, 77 (52.4%) cases were diffuse type, 51 (34.7%) cases were intestinal type and 19 (12.9%) cases were mixed type. For the T stage, 12 (8.2%) cases were in the T3 stage, and 135 (91.8%) were in the T4 stage. For the N stage, 22 (15.0%) cases were in the N1 stage, 28 (19.0%) were in the N2 stage, and 97 (66.0%) cases were in the N3 stage. There were 51 (34.7%) cases diagnosed with the P1 stage and 96 (65.3%) cases diagnosed with P2 and P3 stages at the time of enrollment. The vast majority of patients (75.5%) received chemotherapy after diagnosis.

### 3.2. PD-L1 Expression

The PD-L1 CPS ranged from 0 to 60, with a mean CPS of 3.84. PD-L1 expression with CPS ≥ 1 and CPS-negative was detected in 67 (45.58%) and 80 (54.42%) patients, respectively. High PD-L1 expression at PD-L1 CPS ≥ 10 was identified in 21 (14.29%) patients (Figure 1). Patient baseline characteristics with respect to PD-L1 expression are shown in Table 1. Grouping was conducted according to PD-L1 expression with CPS ≥ 1 and PD-L1 expression with CPS ≥ 10. As for the Lauren classification, there were more intestinal types in the high PD-L1 expression group. Our data showed that the baseline characteristics included gender, age, ECOG PS, tumor localization, pathological differentiation, primary tumor stage, regional lymph nodes stage, degree of peritoneal metastasis, and postoperative chemotherapy treatment, and they were basically consistent among different groups.

### 3.3. Survival Analysis

For the 147 enrolled gastric cancer patients with peritoneal metastasis, the 1-, 3-, and 5-year overall survivals were 68.0%, 15.8%, and 8.1%, respectively. The median OS was 18.53 (95% CI: 12.76–24.31) months in the CPS negative group and 20.87 (95% CI: 14.91–26.82) months in the CPS ≥ 1 group; the OS difference between the two groups was not significant (*p* = 0.150) (Figure 2A). The median PFS was 12.50 (95% CI: 9.29–15.71) months in the CPS negative group and 15.47 (95% CI: 12.64–18.29) months in the CPS ≥ 1 group; the PFS difference between the two groups was significant (*p* = 0.042) (Figure 2B). Nonetheless, the median OS was 18.53 (95% CI: 14.93–22.13) months in the CPS < 10 group and 27.00 (95% CI: 13.81–40.19) months in the CPS ≥ 10 group; the OS difference between the two groups was significant (*p* = 0.015) (Figure 2C). Meanwhile, the median PFS was 12.83 (95% CI: 9.93–15.73) months in the CPS < 10 group and 26.47 (95% CI: 4.56–48.37) months in the CPS ≥ 10 group; the PFS difference between the two groups was significant (*p* = 0.003) (Figure 2D).

### 3.4. Univariate and Multivariate Analysis

We explored prognostic factors for OS in all the 147 enrolled gastric cancer patients with peritoneal metastasis. In the univariable survival analysis, ECOG PS (HR 2.081, 95% CI: 1.368–3.165, *p* < 0.001) (Figure 3A), peritoneal metastasis (HR 1.550, 95% CI: 1.050–2.288, *p* = 0.026) (Figure 3B), chemotherapy (HR 0.511, 95% CI: 0.335–0.780, *p* = 0.002) (Figure 3C), and high PD-L1 expression (HR 0.490, 95% CI: 0.273–0.881, *p* = 0.015) (Figure 2C) were significantly associated with overall survival. On the other hand, gender (HR 1.441, 95% CI: 0.993–2.093, *p* = 0.055), age (HR 0.861, 95% CI: 0.589–1.258, *p* = 0.439), tumor localization (middle, HR 0.831, 95% CI: 0.514–1.342, and lower, HR 0.667, 95% CI: 0.434–1.026, *p* = 0.177), pathological differentiation (HR 0.982, 95% CI: 0.551–1.751, *p* = 0.951), Lauren classification (intestinal, HR 0.632, 95% CI: 0.423–0.944, and mixed, HR 0.959, 95% CI: 0.537–1.713, *p* = 0.073), primary tumor stage (HR 1.811, 95% CI:0.879–3.728, *p* = 0.107), and regional lymph node stage (N2 stage, HR 1.453, 95% CI: 0.766–2.756, and N3 stage, HR 1.068, 95% CI: 0.623–1.830, *p* = 0.383) were not associated with overall survival.

Then, variables with a *p* value < 0.05 in the univariate analysis were entered into multivariate analysis using Cox proportional hazard regression models. The multivariate analysis, including ECOG PS, peritoneal metastasis, chemotherapy, and high PD-L1 expression, demonstrated that poor ECOG PS (HR 1.972, 95% CI: 1.289–3.015, *p* = 0.002) and severe peritoneal metastasis (HR 1.532, 95% CI: 1.036–2.266, *p* = 0.033) remained significantly associated with poor survival, while chemotherapy (HR 0.514, 95% CI: 0.334–0.791, *p* = 0.002) and high PD-L1 expression (HR 0.446, 95% CI: 0.246–0.809, *p* = 0.008) remained independent and significantly favorable prognostic factors (Table 2).

## 4. Discussion

In this study, we focused on the expression of PD-L1 in gastric cancer with peritoneal metastasis and explored its relationship with prognosis. It has been reported that immunotherapy could improve the prognosis of gastric cancer patients and has been widely applied in clinics. The CheckMate 649 trial demonstrated that the PD-1 inhibitor combined with chemotherapy had better OS, PFS benefits, and an acceptable safety profile compared to chemotherapy alone in previously untreated patients with advanced gastric, gastro-esophageal junction, or esophageal adenocarcinoma. However, only 24% of enrolled patients in both nivolumab plus chemotherapy and chemotherapy alone groups had peritoneal metastasis [18]. The ATTRACTION-4 Phase III study showed that patients receiving PD-1 inhibitor combined with chemotherapy had higher ORR rates, longer DOR times, and more significant improvements in PFS, which is consistent with the CheckMate 649 study. In this study, 46% of enrolled patients presented with peritoneal metastasis [29]. In our study, we focused specifically on gastric cancer patients with peritoneal metastasis. In order to eliminate the influence of difference in immunotherapy efficacy caused by different expressions of PD-L1 on prognosis, we retrospectively selected patients who had not received immunotherapy. Through the immunohistochemistry of pathological specimens in gastric cancer patients with peritoneal metastasis, we found that nearly half of the patients presented with PD-L1 expression; meanwhile, high PD-L1 expression predicted a better prognosis.

Currently, immunohistochemistry is widely applicated to detect PD-L1 expression, and several standardized PD-L1 IHC assays have been developed for predicting responses to PD-1 antibody in clinical studies. In the phase II KEYNOTE-059 trial, Dako 22C3 pharmDx assay was used to evaluate PD-L1 expression [17]. For the phase III ATTRACTION-2 and CheckMate 649 trial, Dako 28-8 pharmDx assay was applied [16,18]. Ahn et al. compared PD-L1 CPS with 22C3 pharmDx assay and 28-8 pharmDx assay in patients with gastric cancer and found that the two assays were highly comparable at various CPS cutoff points [30]. A previous study analyzed PD-L1 expression in gastric cancer patients in China; the PD-L1 expression was found in 759 of 1014 (74.85%) cases, and the majority of them were in early and middle stages, with only 66 (6.51%) patients presenting with metastatic gastric cancer [26]. Another study analyzed PD-L1 expression in gastric cancer of Western patients, and 140 of 465 (30.1%) cases showed a membranous PD-L1 expression in tumor cells [22]. In this study, we used Dako 22C3 pharmDx assay to evaluate the expression of PD-L1 in gastric cancer with peritoneal metastasis. Our data showed that positive PD-L1 expression was detected in 67 (45.58%) patients and high PD-L1 expression was identified in 21 (14.29%) patients.

The prognosis of gastric cancer patients with peritoneal metastasis is poor. Previous studies have shown that ECOG PS, the degree of peritoneal metastasis, palliative surgery, and palliative chemotherapy, are important factors affecting the prognosis [11,31]. Our results demonstrated that poor ECOG PS and severe peritoneal metastasis are significantly associated with poor survival, while palliative chemotherapy is a favorable prognostic factor, which is consistent with other studies. As for the relationship between PD-L1 expression and prognosis, although it has been extensively studied, there is still controversy. Three previous meta-analyses have shown that positive PD-L1 expression is associated with a shorter OS [25,32,33]. However, a recent meta-analysis that included 2298 patients from 11 studies showed no significant association between PD-L1 expression with OS [34]. Conversely, some studies suggested that patients with higher levels of PD-L1 had a significantly better OS [35,36]. The strikingly different results may have something to do with tumor staging and choice of treatment. Our results showed that patients with positive PD-L1 expression had a longer OS than the negative group, but there was no statistical difference. Then, we explored the differences in PFS between the two groups and were surprised to find that patients with positive PD-L1 expression had a longer PFS than the negative group, and there was a statistical difference. Furthermore, our data demonstrated that high PD-L1 expression with CPS ≥ 10 had better OS and PFS than the low group, and there was a statistical difference. Therefore, we hypothesized that more convincing survival analysis results might be obtained if supported by a sufficiently large sample size.

Our data suggest that high expression of PD-L1 is an independent protective prognostic factor. There are several possible reasons. First, the higher expression of PD-L1 indicates more severe immunosuppression. Through palliative resection, the high tumor burden is light, and the immunosuppression induced by the high tumor PD-L1 is partially relieved. Therefore, the body’s anti-tumor immunity may work better. Second, 111 of 147 (75.5%) enrolled patients received fluorouracil and platinum-based palliative chemotherapy, which can induce tumor cell death, especially immunogenic cell death, and the dead cancer cells release antigens, which can be captured by dendritic cells (DCs) and activate anti-tumor immunity [37,38]. Consequently, patients with high PD-L1 expression may benefit more from palliative surgery resection and palliative chemotherapy. Moreover, patients with high PD-L1 expression gastric cancer are more likely to benefit from chemotherapy combined with immunotherapy, which is consistent with clinical practice. In the PD-L1 CPS ≥ 10 group, there were two patients that stood out from all others with longer survival and better outcomes. We specially reviewed the medical records of these two patients and found that they had good ECOG PS status before treatment, and the degree of peritoneal metastasis was relatively mild at the time of enrollment; moreover, they received more than six courses of palliative chemotherapy after surgery and had regular follow-ups. Therefore, patients with gastric cancer and peritoneal metastasis may benefit more from better ECOG PS status, a less severe degree of peritoneal metastasis, a higher expression of PD-L1, and palliative chemotherapy.

We acknowledge that limitations exist in the present study. First, the postoperative tissues and clinicopathologic characteristics were collected in a single center, and we made retrospective analyses of those data. It is hard to avoid the selection bias. Second, since gastric cancer with peritoneal metastasis is rarely treated with palliative gastrectomy unless it is in emergency conditions, the number of enrolled patients in our study was relatively small. In order to obtain a larger sample size, we continued to enroll patients from 2000 to 2015 to equalize baseline characteristics of the included variables among different groups. Third, this clinical study mainly focused on patients with peritoneal metastasis that underwent palliative gastrectomy. For patients with peritoneal metastasis who did not undergo palliative gastrectomy, there was no further elaboration. Finally, the retrospective design of this study introduces the potential for selection bias and may impair the ability to establish the relationship between PD-L1 expression and prognosis. In the future, more studies are needed to confirm the results.

## 5. Conclusions

The present study mainly focused on the expression of PD-L1 in gastric cancer with peritoneal metastasis. Our data demonstrated that PD-L1 expression was widely presented in gastric cancer patients with peritoneal metastasis, and a combined positive score (CPS) can effectively evaluate the expression of PD-L1, while a CPS no less than 10 predicts better prognosis.

## Figures and Tables

**Figure 1 biomedicines-11-02003-f001:**
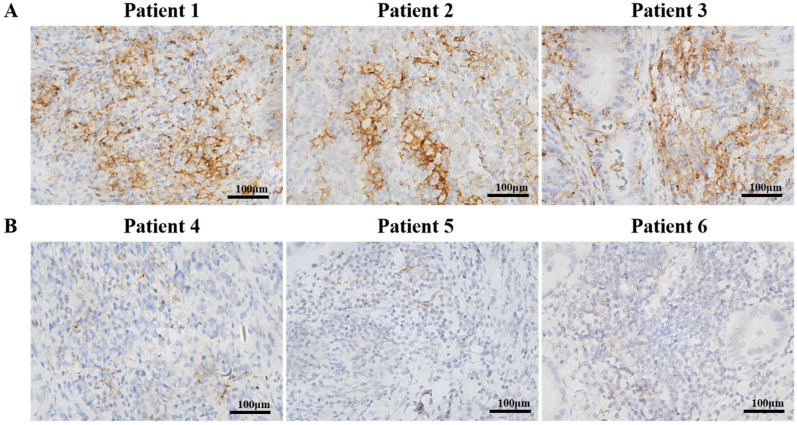
Immunohistochemistry analysis of the expression of PD-L1 in gastric cancer patients with peritoneal metastasis. (**A**) Representative of tissue sections of patients with high PD-L1 expression. (**B**) Representative of tissue sections of patients with low PD-L1 expression. The scale bar represents 100 μm.

**Figure 2 biomedicines-11-02003-f002:**
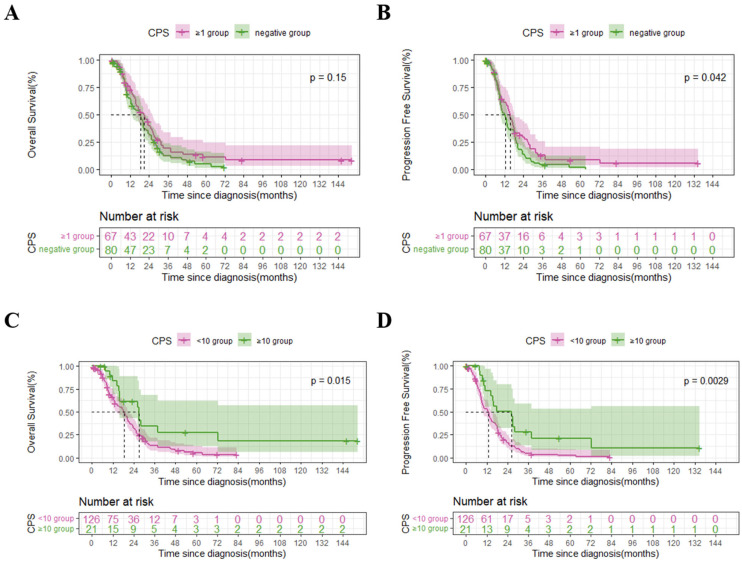
Comparison of overall survival (OS) and progression-free survival (PFS) for the ex-pression of PD-L1 in the 147 gastric cancer patients with peritoneal metastasis. The dashed lines indicate the median OS in (**A** and **C**) and the median PFS in (**B** and **D**). (**A**) The OS of the positive PD-L1 expression group. (**B**) The PFS of the positive PD-L1 expression group. (**C**) The OS of the high PD-L1 expression group. (**D**) The PFS of the high PD-L1 expression group.

**Figure 3 biomedicines-11-02003-f003:**
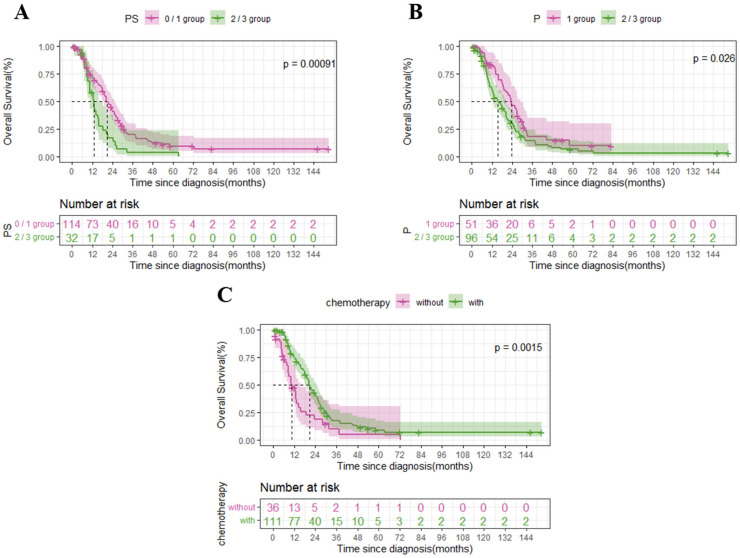
Comparison of overall survival (OS) for the ECOG PS, degree of peritoneal metastasis, and chemotherapy in the 147 gastric cancer patients with peritoneal metastasis. The dashed lines indicate the median OS. (**A**) For ECOG PS. (**B**) For degree of peritoneal metastasis. (**C**) For chemotherapy.

**Table 1 biomedicines-11-02003-t001:** Clinicopathological features according to the PD-L1 expression in gastric cancer patients with peritoneal metastasis.

		PD-L1 Expression			PD-L1 Expression		
	*n* (%)	CPS ≥ 1	CPS-Negative	*p*-Value	CPS ≥ 10	CPS < 10	*p*-Value
Gender	147			0.244			0.270
female	58 (39.5%)	23	35		6	52	
male	89 (60.5%)	44	45		15	74	
Age	147			0.262			0.670
<60	97 (66.0%)	41	56		13	84	
≥60	50 (34.0%)	26	24		8	42	
ECOG PS	147			0.680			0.872
0/1	114 (77.6%)	53	61		16	98	
2/3	33 (22.4%)	14	19		5	28	
Localization	147			0.195			0.957
Upper	45 (30.6%)	17	28		7	38	
Middle	37 (25.2%)	15	22		5	32	
Lower	65 (44.2%)	35	30		9	56	
Differentiation	147			0.315			0.451
Moderately	15 (10.2%)	5	10		3	12	
Poorly	132 (89.8%)	62	70		18	114	
Lauren type	147			0.210			0.044
Diffuse	77 (52.4%)	30	47		6	71	
Intestinal	51 (34.7%)	26	26		12	39	
Mixed	19 (12.9%)	11	8		3	16	
T stage	147			0.748			1.000
T3	12 (8.2%)	6	6		1	11	
T4	135 (91.8%)	61	74		20	115	
N stage	147			0.124			0.351
N1	22 (15.0%)	7	15		1	21	
N2	28 (19.0%)	10	18		4	24	
N3	97 (66.0%)	50	47		16	81	
Peritoneal metastasis	147			0.932			0.396
P1	51 (34.7%)	23	28		9	42	
P2/3	96 (65.3%)	44	52		12	84	
Chemotherapy	147			0.820			0.638
No	36 (24.5%)	17	19		6	30	
Yes	111 (75.5%)	50	61		15	96	

**Table 2 biomedicines-11-02003-t002:** Univariate and multivariate analysis of overall survival in gastric cancer patients with peritoneal metastasis.

Variables	Univariate Analysis		Multivariate Analysis	
	HR (95% CI)	*p* Value	HR (95% CI)	*p* Value
Gender		0.055		
Male	1			
Female	1.441 (0.993–2.093)			
Age (years)		0.439		
<60	1			
≥60	0.861 (0.589–1.258)			
ECOG PS		<0.001		0.002
0/1	1		1	
2/3	2.081 (1.368–3.165)		1.972 (1.289–3.015)	
Localization		0.177		
Upper	1			
Middle	0.831 (0.514–1.342)			
Lower	0.667 (0.434–1.026)			
Differentiation		0.951		
Poorly	1			
Moderately	0.982 (0.551–1.751)			
Lauren type		0.073		
Diffuse	1			
Intestinal	0.632 (0.423–0.944)			
Mixed	0.959 (0.537–1.713)			
T stage		0.107		
T3	1			
T4	1.811 (0.879–3.728)			
N stage		0.383		
N1	1			
N2	1.453(0.766–2.756)			
N3	1.068 (0.623–1.830)			
Peritoneal metastasis		0.026		0.033
P1	1		1	
P2/3	1.550 (1.050–2.288)		1.532 (1.036–2.266)	
Chemotherapy		0.002		0.002
No	1		1	
Yes	0.511 (0.335–0.780)		0.514 (0.334–0.791)	
PD-L1 expression		0.150		
CPS-negative	1			
CPS ≥ 1	0.765 (0.530–1.103)			
PD-L1 expression		0.015		0.008
CPS < 10	1		1	
CPS ≥ 10	0.490 (0.273–0.881)		0.446 (0.246–0.809)	

## Data Availability

All data used for analysis are presented in the tables in this article.

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
