# Peer review of "Prognostic Significance of PD-L1 Expression in Gastric Cancer Patients with Peritoneal Metastasis"

_biomedicines, 2023, doi:10.3390/biomedicines11072003_

Round 1

Reviewer 1 Report

This study aims to investigate PD-L1 expression in patients with peritoneal metastasis in gastric cancer. The study indicates that PD-L1 expression is prevalent in these patients, and higher PD-L1 expression, specifically CPS≥10, is associated with a better prognosis. However, there are some major issues that need further attention and clarification.

Major concerns:

1. In the 3.3. Survival analysis AND Table 2.

The HR of variables seems to be mismatched in the Table 2, particularly in the variables of ECOG PS (0/1 vs 2/3; HR 1 vs 0.481 [0.316-.731]), Peritoneal metastasis (P1 vs P2/3; HR 1 vs 0.645 [0.437-0.953]), Chemotherapy (No vs Yes; HR 1 vs 1.957 [1.282-2.989]) in the Univariate analysis.

The inconsistency in HR values for variables is also observed in the multivariable analysis.

It is important to thoroughly review the HR values of the variables across the figures, tables, and results presented in this manuscript.

2. Retrospective design, particularly the selection bias of adjuvant chemotherapy.

The retrospective design of this study introduces the potential for selection bias and restricts the ability to establish causal relationships between PD-L1 expression and prognosis.

In particular, the inclusion or exclusion of patients who received adjuvant chemotherapy can influence overall survival (OS) outcomes.

Therefore, it is necessary to provide a more detailed explanation of the criteria for chemotherapy administration in the Materials and Methods section.

Furthermore, a thorough discussion on the role of adjuvant chemotherapy and its potential for introducing selection biases should be included in this study.

3. Pathological features such as lymphovascular invasion, perineural invasion, and lymph node positivity have a significant impact on OS in patients with gastric cancer.

Please further analyze lymphovascular invasion, perineural invasion and lymph nodes positive status in the Univariate analysis and Multivariable analysis.

At least, discussion these pathological features in the Discussion part.

Author Response

Dear Reviewer

We are profoundly grateful for the opportunity given to us for the chance to revise our manuscript entitled “Prognostic Significance of PD-L1 Expression in Gastric Cancer Patients with Peritoneal Metastasis (Manuscript ID: biomedicines-2346275)”.

We have carefully revised the manuscript according to your comment to the best our ability, and we also provide a detailed point-to-point response following this cover letter. We hope that this manuscript could be deemed suitable for publication in Biomedicines.

Please do not hesitate to contact us in case of any inquiries.

Best regards,

Run-Cong Nie, Yong-Ming Chen

Point-to-point response to Reviewer’s Comments

Comments and Suggestions:

  1. In the 3.3. Survival analysis AND Table 2.

The HR of variables seems to be mismatched in the Table 2, particularly in the variables of ECOG PS (0/1 vs 2/3; HR 1 vs 0.481 [0.316-.731]), Peritoneal metastasis (P1 vs P2/3; HR 1 vs 0.645 [0.437-0.953]), Chemotherapy (No vs Yes; HR 1 vs 1.957 [1.282-2.989]) in the Univariate analysis.

The inconsistency in HR values for variables is also observed in the multivariable analysis.

It is important to thoroughly review the HR values of the variables across the figures, tables, and results presented in this manuscript.

Response

Many thanks to the reviewer for careful review. The HR of the variables in Table 2 do not match. In response, we have reprocessed the data and thoroughly reviewed the HR values of the variables across the figures, tables and results presented in our manuscript.

  1. Retrospective design, particularly the selection bias of adjuvant chemotherapy.

The retrospective design of this study introduces the potential for selection bias and restricts the ability to establish causal relationships between PD-L1 expression and prognosis.

In particular, the inclusion or exclusion of patients who received adjuvant chemotherapy can influence overall survival (OS) outcomes.

Therefore, it is necessary to provide a more detailed explanation of the criteria for chemotherapy administration in the Materials and Methods section.

Furthermore, a thorough discussion on the role of adjuvant chemotherapy and its potential for introducing selection biases should be included in this study.

Response

Thanks for the reviewer's suggestion. Selection bias is a problem that all retrospective studies have to face. In order to reduce the impact of postoperative chemotherapy on OS, we included patients who underwent or did not undergo postoperative chemotherapy as the study subjects. Consistent with expectations and most previous studies, postoperative palliative chemotherapy played a positive role in prolonging OS.

Although we routinely recommended palliative chemotherapy for patients who meet the conditions, whether a patient had received chemotherapy was mainly determined by their own subjective judgment, combined with their physical and economic conditions. We have included as many patients as possible to alleviate this bias. For palliative chemotherapy, the prescribed regimen was unified as fluorouracil in combination with platinum-based chemotherapy. The dose prescribed to each patient depended on the treating oncologists but still adhered to the standard National Comprehensive Cancer Network (NCCN) guidelines for gastric cancer clinical practice at the time of treatment. In response, we modified the methodological section and added the limitations of this study.

  1. Pathological features such as lymphovascular invasion, perineural invasion, and lymph node positivity have a significant impact on OS in patients with gastric cancer.

Please further analyze lymphovascular invasion, perineural invasion and lymph nodes positive status in the Univariate analysis and Multivariable analysis.

At least, discussion these pathological features in the Discussion part.

Response

Thanks for the reviewer's suggestion. In the revision, we did more analysis of pathological features, such as T stage and N stage. However, for gastric cancer patients with peritoneal metastasis, neither T stage nor N stage has a significant impact on prognosis. The possible reason is that, according to American Joint Committee on Cancer (AJCC) for Carcinoma of the Stomach (8th ed., 2017), once there is peritoneal metastasis for those patients, it is defined as M1 and stage IV. In this case, compared with distant metastasis, local pathological features of tumors, such as lymphovascular invasion, perineural invasion, and lymph node positivity may have much less impact on OS. Moreover, for all patients enrolled, they were diagnosed with T3 and T4 stage and positive lymph node metastasis.

Reviewer 2 Report

This research article by Chen, et al examining and correlating the expression level of PD-L1 in metastatic gastric cancer patients to the prognosis of cancers. The overall research design is appropriate, and the results shown in this study may provide some important insights to future treatment for metastatic gastric cancer patients. However, there are some major critiques that the authors should address.

Major concerns:
1. The results were inconsistent between the figures and the descriptions. The authors should verify which one was correct.
For example,
Line 180-182
“Nonetheless, the median OS was 18.53 (95%CI: 14.93-22.13) months in CPS<10 group and 27.00 (95%CI: 13.81-40.19) months in CPS≥10 group, the OS difference between the two groups was significant (P=0.015) (Figure 2C).”
However, when you looked into the Figure 2C, the results were completely opposite that CPS >10 (red curve) had a median OS of 18.53 months and CPS <10 (green curve) had better OS (27 months).
The same issue was identified with Fig 2D
Line 182-185
“Meanwhile, the median PFS was 12.83 (95%CI: 9.93-15.73) months in CPS<10 group and 26.47 (95%CI: 4.56-48.37) months in CPS≥10 group, the PFS difference between the two groups was significant (P=0.003) (Figure 2D).”
When you looked at Fig2D, the CPS<10 group (the green curve) had PFS of 26.47 months, while the CPS >10 group (the red curve) had worse PFS (12.83 months).

2. The sample size would be a major concern even that it was mentioned in the discussion by authors. In the CPS high group (>10), there were 2 patients standing out from all others with longer survival and better outcomes. It is not known whether there will be around the same percentage of GC patients showing the same high PD-L1 results if a large cohort of patients is enrolled. With removal of these two patients from this study, the overall survival rate and progression free survival would look no difference in all categories. Perhaps the authors can check literature to see if there are any studies showing that about 10% (2 out of 21 patients in current study) of GC patients with peritoneal metastasis and gastrectomy exhibit similarly high PD-L1 expression.

3. It would be better to include a paragraph in the discussion to talk about the discrepancies of PD-L1/PD-1 expression in the prognosis of GC patients from the literature by comparing your results.

4. I don’t quite follow the saying from line 267-269. 75.5% of patients enrolled in this study received chemotherapy but no immunotherapy. Among these patients, higher PD-L1 expression predicts better outcomes. From that saying, what would happen if immunotherapy targeting PD-1/PD-L1 is given to neutralize or to reduce the expression of PD-L1 in these patients? Would the decrease/blockage of PD-L1 predict better or worse outcome if PD-L1 expression is an independent factor of better prognosis.

5. Please include the sample sources for immunohistochemistry staining in the material section. Are the PD-L1 expression examined from tissue sections of removed tumors (through gastrectomy) and thus referred as the PD-L1 expression in the primary tumors? Are there any samples collected from peritoneal metastatic tissues?

There are a few minor concerns on spelling or incomplete sentences:
Line 79:
But the relationship between expression of PD-L1 and prognostic survival in gastric cancer patients with peritoneal metastasis was unknown.
The relationship between expression of PD-L1 and prognostic survival in gastric cancer patients with peritoneal metastasis was unknown.

Line 137:
Statistical significance was set at P<0.05. And the statistical analyses were performed using R version 4.2.2.
Statistical significance was set at P<0.05, and the statistical analyses were performed using R version 4.2.2.

Line 221-224:
In order to eliminate the difference of immunotherapy efficacy caused by different expression of PD-L1, thus affecting the prognosis. We retrospectively selected patients who had not received immunotherapy.
In order to eliminate the difference of immunotherapy efficacy caused by different expression of PD-L1 thus affecting the prognosis, we retrospectively selected patients who had not received immunotherapy.

Line 255:
statis-tical
statistical

Author Response

Dear Reviewer

We are profoundly grateful for the opportunity given to us for the chance to revise our manuscript entitled “Prognostic Significance of PD-L1 Expression in Gastric Cancer Patients with Peritoneal Metastasis (Manuscript ID: biomedicines-2346275)”.

We have carefully revised the manuscript according to your comment to the best our ability, and we also provide a detailed point-to-point response following this cover letter. We hope that this manuscript could be deemed suitable for publication in Biomedicines.

Please do not hesitate to contact us in case of any inquiries.

Best regards,

Run-Cong Nie, Yong-Ming Chen

Point-to-point response to Reviewer’s Comments

Comments and Suggestions:

  1. The results were inconsistent between the figures and the descriptions. The authors should verify which one was correct.

For example,

Line 180-182

“Nonetheless, the median OS was 18.53 (95%CI: 14.93-22.13) months in CPS<10 group and 27.00 (95%CI: 13.81-40.19) months in CPS≥10 group, the OS difference between the two groups was significant (P=0.015) (Figure 2C).”

However, when you looked into the Figure 2C, the results were completely opposite that CPS >10 (red curve) had a median OS of 18.53 months and CPS <10 (green curve) had better OS (27 months).

The same issue was identified with Fig 2D

Line 182-185

“Meanwhile, the median PFS was 12.83 (95%CI: 9.93-15.73) months in CPS<10 group and 26.47 (95%CI: 4.56-48.37) months in CPS≥10 group, the PFS difference between the two groups was significant (P=0.003) (Figure 2D).”

When you looked at Fig2D, the CPS<10 group (the green curve) had PFS of 26.47 months, while the CPS >10 group (the red curve) had worse PFS (12.83 months).

Response

Many thanks to the reviewer for careful review. The results were inconsistent between the figures(Figure 2C and Figure 2D) and the descriptions. In response, we have reprocessed the data and updated figure2 in our manuscript.

  1. The sample size would be a major concern even that it was mentioned in the discussion by authors. In the CPS high group (>10), there were 2 patients standing out from all others with longer survival and better outcomes. It is not known whether there will be around the same percentage of GC patients showing the same high PD-L1 results if a large cohort of patients is enrolled. With removal of these two patients from this study, the overall survival rate and progression free survival would look no difference in all categories. Perhaps the authors can check literature to see if there are any studies showing that about 10% (2 out of 21 patients in current study) of GC patients with peritoneal metastasis and gastrectomy exhibit similarly high PD-L1 expression.

Response

The number of gastric cancer patients with peritoneal metastasis received palliative gastrectomy and CPS≥10 is relatively small, making it difficult to expand the sample size in the short term. And unfortunately, there is no relevant literature report at present. With a small sample size, we conducted an exploratory study on the prognostic significance of high expression of PD-L1 in gastric cancer with peritoneal metastasis. As previously reported, the median overall survival (OS) after palliative gastrectomy for gastric cancer with peritoneal metastasis was only 11.87-13.1 months[1,2]. In our study, the median OS and PFS in the CPS≥10 group were both over 24 months, much longer than 13.1 months, so we have reason to believe that a CPS≥10 for PD-L1 expression predicts better prognosis. We also explained this in our discussion.

References:

[1] Nie RC, Chen S, Yuan SQ, et al. Significant Role of Palliative Gastrectomy in Selective Gastric Cancer Patients with Peritoneal Dissemination: A Propensity Score Matching Analysis. Ann Surg Oncol. 2016;23(12):3956-3963.

[2] Tokunaga M, Terashima M, Tanizawa Y, et al. Survival benefit of palliative gastrectomy in gastric cancer patients with peritoneal metastasis. World J Surg. 2012;36(11):2637-2643.

  1. It would be better to include a paragraph in the discussion to talk about the discrepancies of PD-L1/PD-1 expression in the prognosis of GC patients from the literature by comparing your results.

Response

Thanks for the reviewer's suggestion. We have included a paragraph in the discussion to talk about the discrepancies of PD-L1/PD-1 expression in the prognosis of GC patients from the literature by comparing our results.

  1. I don’t quite follow the saying from line 267-269. 75.5% of patients enrolled in this study received chemotherapy but no immunotherapy. Among these patients, higher PD-L1 expression predicts better outcomes. From that saying, what would happen if immunotherapy targeting PD-1/PD-L1 is given to neutralize or to reduce the expression of PD-L1 in these patients? Would the decrease/blockage of PD-L1 predict better or worse outcome if PD-L1 expression is an independent factor of better prognosis.

Response

In this study, the patients were mainly enrolled before 2015. Immunotherapy is still in the stage of research and exploration and has not been widely used clinically. Many studies have shown that patients with high PD-L1 standards are more likely to benefit from immunotherapy. In order to eliminate the influence of difference in immunotherapy efficacy caused by different expression of PD-L1 on prognosis, we retrospectively selected patients who had not received immunotherapy. For patients with gastric cancer with peritoneal metastasis who have received surgical treatment, whether immunotherapy can achieve better efficacy is also worthy of further study.

  1. Please include the sample sources for immunohistochemistry staining in the material section. Are the PD-L1 expression examined from tissue sections of removed tumors (through gastrectomy) and thus referred as the PD-L1 expression in the primary tumors? Are there any samples collected from peritoneal metastatic tissues?

Response

Many thanks for the reviewer's suggestion. We performed immunohistochemical staining of PD-L1 expression on tissue sections of primary tumor specimens removed by gastrectomy. We have described this in more detail in the corresponding section of the method.

  1. Comments on the Quality of English Language

There are a few minor concerns on spelling or incomplete sentences:

Line 79:

But the relationship between expression of PD-L1 and prognostic survival in gastric cancer patients with peritoneal metastasis was unknown.

The relationship between expression of PD-L1 and prognostic survival in gastric cancer patients with peritoneal metastasis was unknown.

Line 137:

Statistical significance was set at P<0.05. And the statistical analyses were performed using R version 4.2.2.

Statistical significance was set at P<0.05, and the statistical analyses were performed using R version 4.2.2.

Line 221-224:

In order to eliminate the difference of immunotherapy efficacy caused by different expression of PD-L1, thus affecting the prognosis. We retrospectively selected patients who had not received immunotherapy.

In order to eliminate the difference of immunotherapy efficacy caused by different expression of PD-L1 thus affecting the prognosis, we retrospectively selected patients who had not received immunotherapy.

Line 255:

statis-tical

Statistical

Response

Many thanks to the reviewer for careful review. We have modified the paper in terms of the quality of English language.

Round 2

Reviewer 1 Report

The revised manuscript has been improved.

However, In the Multivariate analysis of the Table 2, the HR of the ECOG PS is still mismatch: ECOG 0/1 vs 2/3 ; HR 1 vs 0.507 (0.332-0.776). Please check again.

Author Response

Dear Reviewer

Once again we are profoundly grateful for the opportunity given to us for the chance to revise our manuscript entitled “Prognostic Significance of PD-L1 Expression in Gastric Cancer Patients with Peritoneal Metastasis (Manuscript ID: biomedicines-2346275)”.

We have revised the manuscript according to your careful review, and we also provide a detailed point-to-point response following this cover letter. We hope that this manuscript could be deemed suitable for publication in Biomedicines.

Please do not hesitate to contact us in case of any inquiries.

Best regards,

Run-Cong Nie, Yong-Ming Chen

Point-to-point response to Reviewer’s Comments

Comments and Suggestions:

However, In the Multivariate analysis of the Table 2, the HR of the ECOG PS is still mismatch: ECOG 0/1 vs 2/3 ; HR 1 vs 0.507 (0.332-0.776). Please check again.

Response

Many thanks to your careful review. In the Multivariate analysis of the Table 2, the HR of the ECOG PS is still mismatch: ECOG 0/1 vs 2/3 ; HR 1 vs 0.507 (0.332-0.776). In response, we have reprocessed the data and reviewed the HR. The HR of the ECOG PS should be: ECOG 0/1 vs 2/3 ; HR 1 vs 1.972(1.289-3.015). We have made corrections in the manuscript (Please see Page 9,Table 2).

Reviewer 2 Report

The revised manuscript addressed all my concerns and I have no reservation to recommend it for publication.

Author Response

Dear Reviewer

We are profoundly grateful for your review and recognition of our manuscript entitled “Prognostic Significance of PD-L1 Expression in Gastric Cancer Patients with Peritoneal Metastasis (Manuscript ID: biomedicines-2346275)”. We hope that this manuscript could be deemed suitable for publication in Biomedicines.

Please do not hesitate to contact us in case of any inquiries.

Best regards,

Run-Cong Nie, Yong-Ming Chen